# Delayed Anastomotic Occlusion after Direct Revascularization in Adult Hemorrhagic Moyamoya Disease

**DOI:** 10.3390/brainsci11050536

**Published:** 2021-04-24

**Authors:** Yu Chen, Fa Lin, De-Bin Yan, He-Ze Han, Ya-Hui Zhao, Li Ma, Yong-Gang Ma, Long Ma, Xun Ye, Rong Wang, Xiao-Lin Chen, Dong Zhang, Yuan-Li Zhao, Shuai Kang

**Affiliations:** 1Department of Neurosurgery, Beijing Tiantan Hospital, Capital Medical University, Beijing 100070, China; chenyu_tiantan@126.com (Y.C.); linfa@ccmu.edu.cn (F.L.); de_bin@163.com (D.-B.Y.); hanheze98@163.com (H.-Z.H.); mushroomheady@126.com (Y.-H.Z.); marygl@hotmail.com (L.M.); mayonggang12345@163.com (Y.-G.M.); mlneurosurgeon@163.com (L.M.); yexun79@hotmail.com (X.Y.); ronger090614@126.com (R.W.); xiaolinchen488@hotmail.com (X.-L.C.); zhangdong0660@aliyun.com (D.Z.); zhaoyuanli301@163.com (Y.-L.Z.); 2Department of Neurosurgery, Peking University International Hospital, Peking University, Beijing 102206, China; 3China National Clinical Research Center for Neurological Diseases, Beijing 100070, China; 4Stroke Center, Beijing Institute for Brain Disorders, Beijing 100069, China; 5Beijing Key Laboratory of Translational Medicine for Cerebrovascular Disease, Beijing 100070, China; 6Beijing Translational Engineering Enter for 3D Printer in Clinical Neuroscience, Beijing 100070, China

**Keywords:** hemorrhagic moyamoya disease, graft patency, delayed anastomotic occlusion, collateral circulation, intracranial perfusion, outcomes

## Abstract

Delayed anastomotic occlusion occurred in a considerable proportion of hemorrhagic moyamoya disease (MMD) patients undergoing direct revascularization. This study aimed to investigate the predictors and outcomes of delayed anastomotic occlusion in adult hemorrhagic MMD. The authors retrospectively reviewed 87 adult hemorrhagic MMD patients. Univariate and multivariate logistic regression analyses were performed. After an average of 9.1 ± 6.9 months of angiographic follow-up, the long-term graft patency rates were 79.8%. The occluded group had significantly worse angiogenesis than the non-occluded group (*p* < 0.001). However, the improvement of dilated anterior choroidal artery–posterior communicating artery was similar (*p* = 0.090). After an average of 4.0 ± 2.5 years of clinical follow-up, the neurological statues and postoperative annualized rupture risk were similar between the occluded and non-occluded groups (*p* = 0.750; *p* = 0.679; respectively). In the multivariate logistic regression analysis, collateral circulation Grade III (OR, 4.772; 95% CI, 1.184–19.230; *p* = 0.028) and preoperative computed tomography perfusion (CTP) Grade I–II (OR, 4.129; 95% CI, 1.294–13.175; *p* = 0.017) were independent predictors of delayed anastomotic occlusion. Delayed anastomotic occlusion in adult hemorrhagic MMD might be a benign phenomenon. Good collateral circulation (Grade III) and compensable preoperative intracranial perfusion (CTP Grade I–II) are independent predictors for this phenomenon. Moreover, the delayed anastomotic occlusion has no significant correlations with the long-term angiographic and neurological outcomes, except neoangiogenesis.

## 1. Introduction

Moyamoya disease (MMD) is characterized by the progressive occlusion of the bilateral distal internal carotid arteries (ICA) or proximal middle cerebral artery (MCA) [1]. Hemorrhagic presentation was less common than intracranial ischemia in the manifestation of MMD, but the hematoma is the leading cause of death in MMD [2]. Recently, direct or combined revascularization has been recognized as the most effective treatment strategy to reduce the risk of future intracranial hemorrhages in hemorrhagic MMD [3,4,5,6,7,8,9,10,11]. However, Zhao et al. and Ge et al., from our institution, demonstrated that hemorrhagic presentation predicts lower patency rates (54.9–77.5%) and unsatisfactory neoangiogenesis after direct/combined bypass [12,13]. Such a high rate of delayed anastomotic occlusion prompted us to conduct an in-depth analysis of this phenomenon. We conducted a multicenter retrospective study to investigate the predictors, pathogenesis, and long-term outcomes of delayed anastomotic occlusion in adult hemorrhagic MMD.

## 2. Methods

### 2.1. Study Design and Participants

The patients included in this study were from a multicenter retrospective MMD database between January 2009 and October 2017. This study protocol was performed according to the guidelines of the Declaration of Helsinki and was approved by the institutional ethics committee. The inclusion criteria were as follows: (1) Patients with a diagnosis of MMD by digital subtraction angiography (DSA) based on the 2012 criteria of the Research Committee on the Pathology and Treatment of Spontaneous Occlusion of the Circle of Willis [14]; (2) Patients with hemorrhagic presentation; (3) Patient age > 18 years; (4) Patients who had undergone direct revascularization. Exclusion criteria were as follows: (1) Diagnosis of moyamoya syndrome that had identified causes; (2) Patients without preoperative DSA or follow-up DSA; and (3) Patients lost to clinical follow-up.

### 2.2. Data Collection and Radiological Assessment

Patient baseline demographic data, clinical features, and imaging data were collected. The rupture risk was represented by annualized hemorrhagic rate, and the observational interval of natural history was defined as the interval between the first diagnosis and the revascularization. In addition, for patients who underwent bilateral revascularization surgery, the duration of follow-up for each hemisphere was calculated separately. The neurological status was evaluated by the modified Rankin Scale (mRS), and mRS > 2 was defined as neurological disability. The Suzuki stage system, dilation of anterior choroidal artery-posterior communicating artery (AChA-PCoA), posterior cerebral artery (PCA) involvement, collateral circulation stage system, and preoperative computed tomography perfusion (CTP) stage were used to evaluate the severity of MMD [1,15,16,17]. Collateral circulation was evaluated by the grading system proposed by Liu et al. [15], and we further optimized the 12-point scoring system into a 3-level system: Grade I: 1–4, Grade II: 5–8, Grade III: 9–12. Cerebral perfusion was evaluated by the CTP staging system (pre-infarction staging system) proposed in our previous studies [17,18]. The postoperative neoangiogenesis was classified into three categories according to the criteria proposed by Matsushima: Level A, more than 2/3 of the MCA distribution; Level B, between 2/3 and 1/3 of the MCA distribution; and Level C, slight or no MCA distribution [19].

Clinical follow-up was conducted in the first 3–6 months and annually after surgery by clinical visit or telephone interview. Angiographic follow-up was performed at the first inpatient follow-up, between 3 months and 1 year postoperatively. The evaluation of mRS score was conducted by two neurosurgeons who have at least 5 years’ experience of clinical practice, and a series of training programs were performed to ensure the accuracy of mRS measurement. All the images were interpreted independently by at least two radiologists who had worked more than 5 years in our institute’s radiology center. Researchers who performed follow-up assessments were blinded to the graft patency.

### 2.3. Surgical Procedures and Perioperative Management

The indication for revascularization was based on the guidelines set by the Japanese Ministry of Health and Welfare [14]. The symptomatic, and hemodynamically affected, hemisphere was the preferred side for revascularization [18]. In our institution, direct revascularization involves the end-to-side anastomosis of the branches of the superficial temporal artery (STA) (anterior branch, posterior branch, or both) to the cortical branches of the MCA. The ideal recipient artery is about the same caliber as the donor STA branch and without atherosclerotic changes. Interrupted and running suture anastomosis techniques were randomly applied during the operation. Graft patency was routinely confirmed with intraoperative indocyanine green videoangiography (ICG).

Computed tomography (CT), magnetic resonance imaging (MRI), DSA, and CTP were conducted for all patients, no more than 3 months before revascularization. MRI included T2-weighted and diffusion-weighted imaging (DWI) sequences, which were performed to confirm the cerebral perfusion was stable and no subacute or acute infarction or hemorrhage. A standardized perioperative management protocol was applied to all cases, and aspirin was not recommended in the hemorrhagic MMD patients. The daily intravenous fluid volume was 3000–3500 mL, and the systolic blood pressure was maintained at 120–140 mmHg to keep the normal perfusion pressure. Antiepileptic drugs (sodium valproate, 200 mg twice daily) were routinely used within seven days after surgery. All patients underwent CT scanning on postoperative day 1 to detect whether there was acute infarction or hemorrhage. Repeat CT or MRI was performed in the cases with severe headache or postoperative neurological deterioration at any time. Computed tomography angiography (CTA) and CTP were reexamined 3–7 days after the revascularization.

### 2.4. Statistical Analysis 

Categorical variables were presented as counts (with percentages); continuous variables were presented as the mean ± standard deviations (SD). Patients were divided into occluded and non-occluded groups. In the comparison of baseline characteristics and outcomes, the Pearson chi-square test, Fisher exact test, or Kruskal–Wallis ANOVA test were used to compare categorical variables as appropriate, and the two-tailed *t*-test or one-way ANOVA test were employed to compare normal distribution continuous variables. The Wilcoxon rank sum test was applied to compare non-normal distribution continuous variables. The Poisson rate test was used to compare the differences in annualized rupture risk. Univariate and multivariate logistic regression analyses were used to calculate odds ratios (ORs) and 95% confidence intervals (CI) for predictors of delayed anastomotic occlusion and long-term unfavorable outcomes (mRS > 2). Characteristics with *p* < 0.10 in the univariate regression were subsequently included in multivariate regression analyses, with adjustments for other characteristics. Kaplan–Meier analysis (Log Rank, Mantel–Cox) was employed to compare the hemorrhage-free survival rates between occluded and non-occluded patients. *p* < 0.05 was considered to be statistically significant. Statistical analysis was performed using SPSS (V.25.0, IBM, New York, NY, USA).

## 3. Results

### 3.1. Baseline Characteristics

A total of 87 patients, with 94 operated hemispheres, were included in this study from our multicenter retrospective MMD database of 1333 patients, according to the inclusion criteria (Figure 1). The mean age at the operation was 38.2 ± 10.8 years, and 46.8% were male (Table 1). The preoperative annualized re-rupture risk was 9.3%. In terms of angiographic characteristics, most hemispheres were presented with Suzuki stage III–IV (59.6%), and 66 (70.2%) hemispheres were presented with dilated AChA-PCoA. In total, 53 hemispheres (56.4%) had significant hypoperfusion (CTP stage: Grade III–IV), and 68 hemispheres (72.3%) had moderate collateral circulation compensation (collateral circulation stage: Grade II).

### 3.2. Angiographic and Clinical Outcomes

The intraoperative graft patency rate was confirmed by intraoperative ICG to be 100.0%. However, the perioperative CTA revealed the perioperative graft patency rate was 91.5%. After an average of 9.1 ± 6.9 months of angiographic follow-up, the postoperative DSA demonstrated that in 19 hemispheres (20.2%) delayed anastomotic occlusion occurred, indicating the long-term graft patency rate of 79.8% (Table 2).

At the last angiographic follow-up, 60 hemispheres (63.8%) achieved satisfactory neoangiogenesis (Matsushima Level A and B), and 80.3% of patients with AChA-PCoA dilation improved after the revascularization. Compared with the non-occluded group, the occluded group had significantly worse angiogenesis (21.1% vs. 74.7%, *p* < 0.001). After an average of 4.0 ± 2.5 years of clinical follow-up, six patients (6.4%) died of intracranial hemorrhage during clinical follow-up, and the long-term neurological disability rate was 14.9%. There were no significant differences in the long-term neurological statues between the occluded group and non-occluded group (*p* = 0.750). In the multivariate logistic regression analysis of unfavorable outcomes (mRS > 2), multiple preoperative hemorrhage (OR, 13.153; 95% CI, 2.587–66.882; *p* = 0.002) and higher Suzuki stage (OR, 2.080; 95% CI, 1.066–4.058; *p* = 0.032) were independent predictors. However, the delayed anastomotic occlusion (OR, 0.280; 95% CI, 0.040–1.981; *p* = 0.202) had no significant correlation with unfavorable outcomes (mRS > 2) (Table 3).

In terms of postoperative rupture risk, 14 postoperative hemorrhagic events occurred in 14 hemispheres (14.9%) (4 in the occluded group and 10 in the non-occluded group, *p* = 0.629) during a total of 378.6 patient-years clinical follow-ups, translating to a postoperative annualized rupture risk of 3.7%. The postoperative annualized rupture risk was significantly decreased compared with the natural annualized re-rupture risk (3.7% vs. 9.3%, *p* = 0.010). The Poisson rate test of annualized rupture risk showed no significant differences between the occluded group and the non-occluded group (4.4% vs. 3.5%, *p* = 0.679). Kaplan–Meier analysis proposed no significant differences in postoperative hemorrhage-free survival between the two groups (*p* = 0.678) (Figure 2).

### 3.3. Predictors of Delayed Anastomotic Occlusion

In the univariate logistic regression analysis, collateral circulation Grade III (OR, 8.192; 95% CI, 2.027–33.105; *p* = 0.003) and preoperative CTP Grade I–II (OR, 4.978; 95% CI, 1.617–15.328; *p* = 0.005) were found to be associated with the delayed anastomotic occlusion (Table 4). Interestingly, perioperative graft patency (OR, 2.700; 95% CI, 0.584–12.474; *p* = 0.396) has no significant correlation with the delayed anastomotic occlusion. In the multivariate logistic regression analysis, collateral circulation Grade III (OR, 4.772; 95% CI, 1.184–19.230; *p* = 0.028) and preoperative CTP Grade I–II (OR, 4.129; 95% CI, 1.294–13.175; *p* = 0.017) were significantly associated with the delayed occluded anastomosis.

## 4. Discussion

Direct revascularization is currently recognized as the superior treatment paradigm for adult hemorrhagic MMD to reduce the incidence of rebleeding [9,10,20]. However, hemorrhagic MMD was indicated to be more prone to delayed anastomotic occlusion compared with ischemic MMD, with a long-term graft patency rate of 54.9–77.5% [12,13]. Such a high rate of anastomotic occlusion prompted us to conduct in-depth research on the mechanism of delayed anastomotic occlusion and its impact on the long-term outcomes. In this study, we conducted a pioneering study on delayed anastomotic occlusion in adult hemorrhagic MMD. We found that the long-term graft patency rate was 79.8% in adult hemorrhagic MMD patients who underwent direct revascularization. Good collateral circulation (collateral circulation Grade III) and compensable preoperative intracranial perfusion (CTP Grade I–II) were independent predictors for delayed anastomotic occlusion. Meanwhile, we found no significant correlation between the delayed anastomotic occlusion and the long-term angiographic and neurological outcomes except neoangiogenesis.

### 4.1. Perioperative and Long-Term Graft Patency

Several previous studies have proposed that the incidence of perioperative anastomotic occlusion in MMD patients undergoing direct bypass procedures, ranged from 4.0% to 35.5% [21,22,23]. Spontaneous recanalization of the occluded anastomosis may occur in many occluded patients (30.4–100.0%) during 3–12 months of postoperative CTA follow-up [21,22]. In this study, perioperative anastomotic occlusion occurred in eight hemispheres, with a perioperative graft patency rate of 91.5%. Spontaneous recanalization was confirmed in five hemispheres (62.5%) with perioperative occluded anastomosis during long-term angiographic follow-up, similar to previous studies. Kim et al. suggested that the pathogenesis of spontaneous recanalization might be the natural thrombolytic activity of anastomotic thrombus, natural recovery of the bypass pedicle into the unfolded form, or detumescence of temporalis muscle [21,24,25].

The long-term graft patency rate after direct or combined revascularization for overall MMD was reported to be 88.3–100.0%, and hemorrhagic presentation was indicated to have an inverse effect on graft patency [12,13,18,21,23,26,27,28]. About 22.5–45.1% direct anastomosis was observed to be occluded during postoperative angiographic follow-up of 10–18 months [12,13]. The present study suggested that the long-term graft patency rate of direct revascularization in adult hemorrhagic MMD was 79.8%, during an average of 9.1 ± 6.9 months of angiographic follow-up, which was slightly higher than in previous studies.

### 4.2. Pathogenesis of Delayed Anastomotic Occlusion

Delayed occlusion represents a potential complication in vascular surgery as a whole; vasospasm, space occupying effect, thrombosis, and altered perfusion may be the underlying mechanisms [29,30,31]. In this study, the perioperative CTA of most patients with delayed anastomotic occlusion (16 of 19, 84.2%) indicates that the perioperative anastomosis is unobstructed, which means that the pathogenesis of delayed anastomotic occlusion may be quite different from that of perioperative anastomotic occlusion. The thrombogenic event might be the main cause of perioperative anastomotic occlusion [21]. However, our previous study indicated that antiplatelet therapy (aspirin) may improve outcomes of ischemic MMD, but does not increase the patency rate of the bypass graft [32].

In previous studies, Yoon et al. speculated that the high demand to augment intracranial blood flow encourages bypass graft patency. Yin et al. found that hemorrhagic MMD patients suffer less from hypoperfusion [17,23]. Amin et al. demonstrated a significant decline in direct bypass flow, compared with baseline by quantitative magnetic resonance angiography, and angiography in these hemispheres demonstrated prominent indirect collaterals [33]. Therefore, the conversion of blood flow between the donor vessel and the recipient vessel may be a potential cause of delayed vascular anastomotic occlusion. The present study recognized that good collateral circulation formation (collateral circulation Grade III) and compensable preoperative intracranial perfusion (CTP Grade I–II) were significantly associated with the delayed occluded anastomosis. Therefore, we hypothesized that the mechanism of delayed anastomotic occlusion in adult hemorrhagic MMD was passive disuse occlusion caused by the relatively adequate intracranial perfusion compensation.

### 4.3. Long-Term Angiographic and Clinical Outcomes

The long-term angiographic and clinical outcomes of delayed anastomotic occlusion in hemorrhagic MMD has not been analyzed in previous studies. In terms of rupture risk, the postoperative rupture risk was reported to be reduced from 3.7–17% per year to about 1.9% per year after direct revascularization [2,8,10,11,34]. In this study we also confirmed this phenomenon (9.3% to 3.7%). However, we found no significant differences between postoperative rupture risk and delayed occluded anastomosis (4.4% vs. 3.5%).

Dilated AChA-PCoA was reported to be present in 41.7–71.9% of hemorrhagic MMD patients [16,35,36,37], and 23.2–70.1% of them would be improved after revascularization [27,34,38]. This change in vascular remodeling may be related to ipsilateral ICA disuse occlusion caused by extracranial-intracranial (EC-IC) blood flow conversion after revascularization [39,40]. In this study, 80.3% of patients observed improvement in dilated AChA-PCoA after direct revascularization, and the improvement of dilated AChA-PCoA has no significant correlation with graft patency. Neoangiogenesis was recognized as the most intuitive angiographic prognostic indicator after revascularization in MMD patients [19,39]. Previous studies reported that 64.8–85.3% of hemorrhagic MMD patients could achieve satisfactory neoangiogenesis (Matsushima Level A, B) after direct/combined bypass [12,13,34]. In this study, we found that 63.8% of adult hemorrhagic MMD patients achieved satisfactory neoangiogenesis, and the non-occluded group could achieve significantly better postoperative neoangiogenesis than the occluded group. Interestingly, 21.1% of the patients with delayed anastomotic occlusion achieved satisfactory neoangiogenesis; the mechanism may be that donor vascular occlusion is a kind of chronic and progressive occlusion, during which neoangiogenesis originates from the fascial tissue on the surface of donor vessel close to the brain surface.

### 4.4. Study Limitations

Several potential limitations of this study should be noted. Firstly, we only included hemorrhagic MMD because we thought the mechanism of delayed anastomotic occlusion was different from ischemic MMD, and it was difficult for us to explore two different types of delayed anastomotic occlusion in one study. However, this resulted in a small sample size and the possible bias from low incidence and accidental events should be fully considered in the predictor analysis. Secondly, we did not include postoperative lifestyle parameters that may cause the delayed anastomotic occlusion, such as postoperative lipid control, postoperative lateral decubitus placement during sleep resulting in the compression of the bypass vessels, etc. Future investigations, with larger sample sizes of hemorrhagic MMD, to explore the delayed anastomotic occlusion and individualized surgical scheme are required.

## 5. Conclusions

The long-term graft patency rate of the direct or combined bypass in adult hemorrhagic MMD was 79.8%. Good collateral circulation (collateral circulation Grade III) and compensable preoperative intracranial perfusion (CTP Grade I–II) are independent predictors for delayed anastomotic occlusion in adult hemorrhagic MMD. Moreover, the graft patency has no significant correlations with the long-term angiographic and neurological outcomes, except neoangiogenesis.

## Figures and Tables

**Figure 1 brainsci-11-00536-f001:**
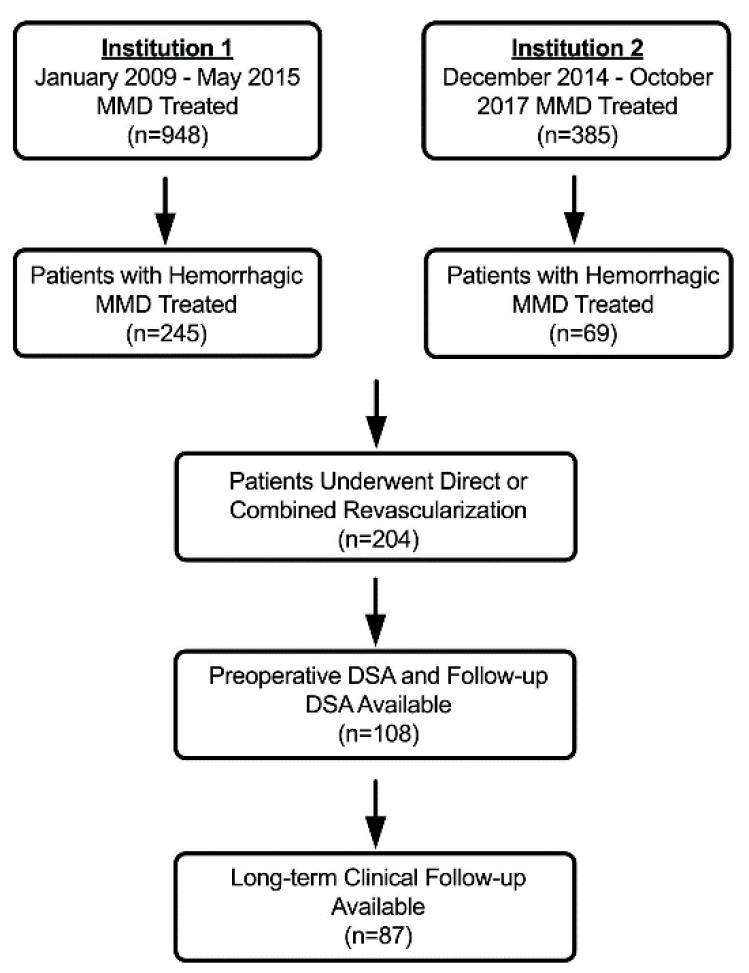
The flow diagram of patient screening.

**Figure 2 brainsci-11-00536-f002:**
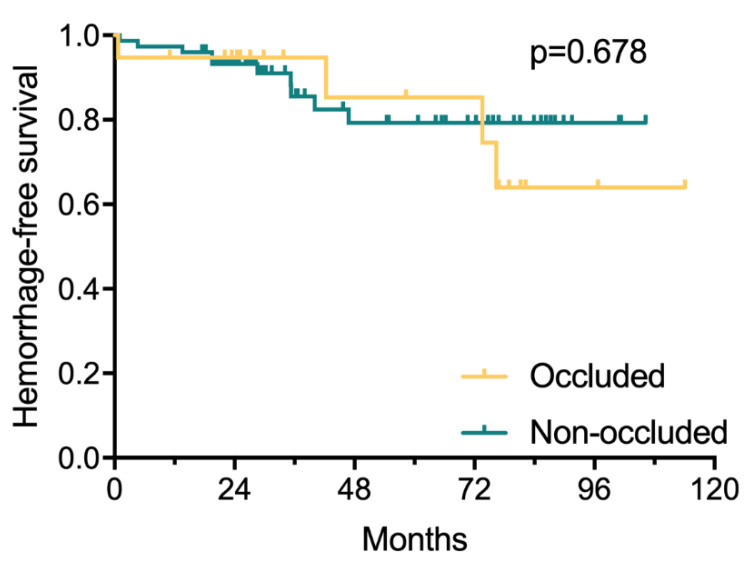
Kaplan–Meier plot showing that there was no significant difference in postoperative hemorrhage-free survival between the occluded and non-occluded group, (*p* = 0.678, log-rank test). The markers in the figure indicate the length of the last follow-up for data from hemorrhage-free patients in the two groups.

**Table 1 brainsci-11-00536-t001:** Baseline characteristics of patients with and without delayed anastomotic occlusion.

Characteristics	All Cases	Occluded	Non-Occluded	*p* Value
No. of patients	87	19	68	
No. of hemispheres	94	19	75	
Age, year	38.2 ± 10.8	40.7 ± 10.5	37.6 ± 10.9	0.260
Sex (male)	44 (46.8%)	6 (31.6%)	38 (50.7%)	0.136
Multiple preoperative hemorrhage	18 (19.1%)	5 (26.3%)	13 (17.3%)	0.574
Total preoperative observation interval, years	192.8	73.9	118.9	
Preoperative annualized re-rupture risk	9.3%	6.8%	10.9%	0.373
History of risk factors				
Hypertension	20 (21.3%)	3 (15.8%)	17 (22.7%)	0.513
Hyperlipidemia	6 (6.4%)	1 (5.3%)	5 (6.7%)	0.823
Smoking	19 (20.2%)	2 (10.5%)	17 (22.7%)	0.227
Preoperative mRS score	1.3 ± 0.6	1.5 ± 0.6	1.3 ± 0.6	0.095
Suzuki stage				
Mean	3.6 ± 1.2	3.8 ± 1.3	3.6 ± 1.2	0.429
I–II	16 (17.0%)	1 (5.3%)	15 (20.0%)	0.131
III–IV	56 (59.6%)	12 (63.2%)	44 (58.7%)	
V–VI	22 (23.4%)	6 (31.6%)	16 (21.3%)	
Dilation of AChA-PCoA	66 (70.2%)	13 (68.4%)	53 (70.7%)	0.228
PCA involvement	23 (24.5%)	5 (26.3%)	18 (24.0%)	0.834
Collateral circulation stage				0.005 *
Grade I (1–4)	15 (16.0%)	1 (5.3%)	14 (18.7%)	
Grade II (5–8)	68 (72.3%)	12 (63.2%)	56 (74.7%)	
Grade III (9–12)	11 (11.7%)	6 (31.6%)	5 (6.7%)	
Preoperative CTP stage				
Mean	2.8 ± 0.8	2.3 ± 0.7	2.9 ± 0.8	0.007 *
I–II	41 (43.6%)	14 (73.7%)	27 (36.0%)	0.003 *
III–IV	53 (56.4%)	5 (26.3%)	48 (64.0%)	
Surgical side (left)	52 (55.3%)	10 (52.6%)	42 (56.0%)	0.878
Anastomosis (interrupted suture)	35 (37.2%)	6 (31.6%)	29 (38.7%)	0.568
Operative duration, hours	3.9 ± 1.2	3.7 ± 0.8	4.0 ± 1.3	0.352
Angiographic follow-up duration, months	9.1 ± 6.9	10.5 ± 7.4	8.7 ± 6.8	0.326
Clinical follow-up duration, years	4.0 ± 2.5	4.8 ± 3.0	3.8 ± 2.3	0.298

AChA = Anterior Choroidal Artery; CTP = Computed Tomography Perfusion; mRS = modified Rankin Scale; PCA = Posterior Cerebral Artery; PCoA = Posterior communicating Artery. In this table, and successive tables, the Suzuki stage, CTP stage, and collateral circulation stage refer to the corresponding stages of the surgical side. Values are expressed as number of cases (%) or mean ± standard deviation, unless otherwise indicated. * Statistical significance (*p* < 0.05).

**Table 2 brainsci-11-00536-t002:** Long-term angiographic and clinical outcomes.

Characteristics	Total	Occluded	Non-Occluded	*p* Value
No. of hemispheres	94	19	75	
Angiographic outcomes				
Matsushima scale (Level A, B)	60 (63.8%)	4 (21.1%)	56 (74.7%)	<0.001 *
Improvement of dilated AChA-PCoA	53 (66, 80.3%)	7 (13, 53.8%)	46 (53, 86.8%)	0.090
Clinical outcomes				
Most recent mRS score	1.2 ± 1.6	1.1 ± 1.5	1.2 ± 1.7	0.750
Neurological disability (mRS > 2)	14 (14.9%)	3 (15.8%)	11 (14.7%)	>0.999
Death	6 (6.4%)	1 (5.3%)	5 (6.7%)	>0.999
No. of postoperative hemorrhage	14 (14.9%)	4 (21.1%)	10 (13.3%)	0.629
Total observation duration, years	378.6	91.2	287.4	
Postoperative annualized rupture risk	3.7%	4.4%	3.5%	0.679

AChA = Anterior Choroidal Artery; mRS = modified Rankin Scale; PCoA = Posterior Communicating Artery. Values are expressed as number of cases (%) or mean ± standard deviation, unless otherwise indicated. Poisson rate test for the preoperative re-rupture risk and the postoperative rupture risk is significant (9.3% vs. 3.7%, *p* = 0.010). Poisson rate test of the annualized rupture risk between patients with and without the delayed anastomotic occlusion is not significant (*p* = 0.679). * Statistical significance (*p* < 0.05).

**Table 3 brainsci-11-00536-t003:** Predictors of unfavorable outcomes (mRS > 2) in hemorrhagic moyamoya disease.

Characteristics	Univariate	*p* Value	Multivariate	*p* Value
	OR (95% CI)		OR (95% CI)	
Age, years	0.990 (0.939–1.043)	0.694		
Sex (male)	1.712 (0.527–5.560)	0.371		
Multiple preoperative hemorrhage	7.000 (2.027–24.179)	0.002 *	13.153 (2.587–66.882)	0.002 *
Preoperative mRS score	2.542 (1.058–6.107)	0.037 *	2.394 (0.861–6.658)	0.094
Suzuki stage	1.806 (1.096–2.978)	0.020 *	2.080 (1.066–4.058)	0.032 *
Dilation of AChA-PCoA	2.295 (1.130–4.660)	0.022 *	2.304 (0.977–5.431)	0.057
PCA involvement	1.284 (0.361–4.567)	0.699		
Collateral circulation stage				
Grade I (1–4)	2.509 (0.668–9.418)	0.173		
Grade II (5–8)	0.600 (0.180–2.001)	0.406		
Grade III (9–12)	0.607 (0.071–5.204)	0.649		
Preoperative CTP stage (I–II)	1.353 (0.434–4.220)	0.602		
Surgical side (left)	1.474 (0.469–4.635)	0.507		
Anastomosis (interrupted suture)	0.382 (0.120–1.213)	0.103		
Operative duration, hours	0.990 (0.977–1.002)	0.109		
Delayed anastomotic occlusion	1.091 (0.272–4.376)	0.902	0.280 (0.040–1.981)	0.202

Multivariate model: Logistic regression model adjusting for the parameters with *p* < 0.1 in the univariate analysis and delayed anastomotic occlusion. AChA = Anterior Choroidal Artery; CTP = Computed Tomography Perfusion; mRS = modified Rankin Scale; PCA = Posterior Cerebral Artery; PCoA = Posterior communicating Artery; CI = Confidence Intervals. Values are expressed as number of cases (%) or mean ± standard deviation, unless otherwise indicated. * Statistical significance (*p* < 0.05).

**Table 4 brainsci-11-00536-t004:** Predictors of delayed anastomotic occlusion in adult hemorrhagic moyamoya disease.

Characteristics	Univariate	*p* Value	Multivariate	*p* Value
	OR (95% CI)		OR (95% CI)	
Age, years	1.028 (0.979–1.080)	0.263		
Sex (male)	2.225 (0.765–6.475)	0.142		
Multiple preoperative hemorrhage	1.875 (0.569–6.183)	0.302		
History of risk factors				
Hypertension	0.640 (0.166–2.459)	0.515		
Hyperlipidemia	0.778 (0.085–7.080)	0.824		
Smoking	0.403 (0.088–1.842)	0.241		
Preoperative mRS score	1.940 (0.877–4.291)	0.102		
Suzuki stage				
Mean	1.184 (0.778–1.801)	0.430		
I–II	0.222 (0.027–1.800)	0.159		
III–IV	1.208 (0.427–3.415)	0.722		
V–VI	1.702 (0.559–5.185)	0.349		
Dilation of AChA-PCoA	1.410 (0.802–2.477)	0.232		
PCA involvement	1.131 (0.358–3.573)	0.834		
Collateral circulation stage				
Grade I (1–4)	0.242 (0.030–1.968)	0.185		
Grade II (5–8)	0.541 (0.185–1.582)	0.262		
Grade III (9–12)	8.192 (2.027–33.105)	0.003 *	4.772 (1.184–19.230)	0.028 *
Preoperative CTP stage				
Mean	0.375 (0.176–0.799)	0.011 *		
I–II	4.978 (1.617–15.328)	0.005 *	4.129 (1.294–13.175)	0.017 *
III–IV	Ref.			
Surgical side (left)	0.924 (0.337–2.533)	0.878		
Anastomosis (interrupted suture)	1.366 (0.467–3.995)	0.569		
Operative duration, hours	0.996 (0.988–1.005)	0.357		
Perioperative graft patency	2.700 (0.584–12.474)	0.396		

Multivariate model: Logistic regression model adjusting for the parameters with *p* < 0.1 in the univariate analysis. AChA = Anterior Choroidal Artery; CI = Confidence Intervals; CTP = Computed Tomography Perfusion; mRS = modified Rankin Scale; OR = Odds Ratios; PCA = Posterior Cerebral Artery; PCoA = Posterior communicating Artery. Values are expressed as number of cases (%) or mean ± standard deviation, unless otherwise indicated. * Statistical significance (*p* < 0.05).

## Data Availability

The data used to support the findings of this study are available from the corresponding author upon reasonable request.

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
