# Peer review of "Delayed Anastomotic Occlusion after Direct Revascularization in Adult Hemorrhagic Moyamoya Disease"

_brainsci, 2021, doi:10.3390/brainsci11050536_

Round 1

Reviewer 1 Report

Line 253-255 (Authors should discuss the results and how they can be interpreted from the per spective of previous studies and of the working hypotheses. The findings and their implications should be discussed in the broadest context possible. Future research directions may also be highlighted.)

Author Response

We thank the Reviewer 1 for his positive comments and constructive suggestions. We have revised the manuscript accordingly. Revisions to the manuscript are tracked.

Reviewer 1

Comment 1: Line 253-255 (Authors should discuss the results and how they can be interpreted from the perspective of previous studies and of the working hypotheses. The findings and their implications should be discussed in the broadest context possible. Future research directions may also be highlighted.)

Authors’ Response : We thank this reviewer for this suggestion. Indeed, the results in this study should be discussed and interpreted from the perspective of previous studies and of the working hypotheses.

In this study, the perioperative graft patency and the spontaneous recanalization rate were similar to previous studies. The thrombogenic event might be the main cause of perioperative anastomotic occlusion, which was discussed in line 257-258. And the underlying pathogenesis of spontaneous recanalization might be the natural thrombolytic activity of anastomotic thrombus, natural recovery of the bypass pedicle into the unfolded form, or detumescence of temporalis muscle, which was discussed in line 241-244.

The long-term graft patency rate was slightly higher than previous studies (79.8% vs. 54.9%-77.5%). However, previous studies just reported the graft patency rate and they have not explored the possible mechanism of the delayed anastomotic occlusion, and we can’t make a hypothesis about the potential mechanism that the long-term graft patency rate in this study is slightly higher than that in previous studies from the review of the previous literature. Therefore, we used the data in this study to analyze the possible mechanism of this delayed anastomotic occlusion. The corresponding discussion is on line 269-276.

Delayed anastomotic occlusion after revascularization in hemorrhagic MMD is a rare phenomenon among rare diseases. Therefore, few studies have been conducted in the past. We have been discussed the findings and implications in the broadest context possible.

Indeed, we should highlight the future research directions after fully analyzing the results of this study. Based on the results of this study, we believe that patients with delayed anastomotic occlusion may not be suitable for direct revascularization procedure. Therefore, we should reconsider the selection of surgical indications and surgical strategies for such patients. Future investigations with larger sample size of hemorrhagic MMD to explore the delayed anastomotic occlusion and individualized surgical scheme are required.

Change to Text : “Future investigations with larger sample size of hemorrhagic MMD to explore the delayed anastomotic occlusion and individualized surgical scheme are required.”

We supplemented the prospects for future research directions in the “Limitations” section of the “Discussion” modules (line 309-311).

Reviewer 2 Report

This paper, whose aim is to investigate the outcomes of delayed anastomotic occlusion in 87 adult hemorrhagic MMD, sounds interesting. Please look at these points:

  1. Table 1: It results that occluded patients, affetting 19 hemispheres, were 19, whereas in some non-occluded patients MMD affected both hemispheres. Does this data create some statistical bias?
  2. Delayed anastomotic occlusion occurred in 79.8% of patients. Did the authors find any additional common risk factors among these patients? In table 4 it seems no, but are there any trend?
  3. English should be revised in some sentences: e.g. "Hemorrhagic presentation was less common than intracranial ischemia as the two main manifestations, but it is the leading cause of death in MMD" ; 
  4. Lines 256-276: "4.2. Pathogenesis of delayed anastomotic occlusion" Unfortunately, delayed occlusion represents a potential complication in vascular surgery as a whole. This point should be discuss more in the section. Please look at these 3 important references.:     Occurrence and impact of delayed cerebral ischemia after coiling and after clipping in the International Subarachnoid Aneurysm Trial (ISAT). J Neurol. 2012 Apr;259(4):679-83. doi: 10.1007/s00415-011-6243-2.      -      Wrapping of intracranial aneurysms: Single-center series and systematic review of the literature. Br J Neurosurg. 2015;29(6):785-91. doi: 10.3109/02688697.2015.1071320.     -      Delayed sudden hearing recovery after treatment of a large vertebral artery aneurysm causing hearing loss and imbalance: a case report. Br J Neurosurg. 2019 Dec 3:1-5. doi: 10.1080/02688697.2019.1698013. 
  5. Lines 297-300: "Interestingly, 21.1% of the patients with delayed anastomotic occlusion achieved satisfactory neoangiogenesis...  the brain surface" Can the authors explain this concept better?

Overall an interesting paper.

Author Response

We thank the Reviewer 2 for his positive comments and constructive suggestions. We have revised the manuscript accordingly. Revisions to the manuscript are tracked.

Reviewer 2

Comment 1: This paper, whose aim is to investigate the outcomes of delayed anastomotic occlusion in 87 adult hemorrhagic MMD, sounds interesting. Please look at these points:

Table 1: It results that occluded patients, affecting 19 hemispheres, were 19, whereas in some non-occluded patients MMD affected both hemispheres. Does this data create some statistical bias?

Authors’ Response : We thank this reviewer for identifying this issue. It may be the inaccurate expression in the text that led to your misunderstanding. In this study, we performed the statistical calculations based on the surgical hemisphere. 80 of 87 hemorrhagic MMD patients underwent unilateral direct revascularization, and 7 patients underwent bilateral direct revascularization (80+7=87, 87+7=94). Therefore, the total number of operating hemispheres is 94. After an average of 9.1±6.9 months angiographic follow-up, the postoperative DSA demonstrated 19 hemispheres in 19 hemorrhagic MMD patients occurred delayed anastomotic occlusion, and the anastomotic sites in the remaining 75 operative hemispheres were unobstructed (19+75=94). So, the non-occluded patients did not affect both hemispheres, they just affect the ipsilateral hemisphere.

Change to Text : None.

Comment 2: Delayed anastomotic occlusion occurred in 79.8% of patients. Did the authors find any additional common risk factors among these patients? In table 4 it seems no, but are there any trend?

Authors’ Response : We thank this reviewer for identifying this issue. It may be the inaccurate expression in the text that led to your misunderstanding. Delayed anastomotic occlusion occurred in 20.2% hemispheres, not in 79.8% of patients. In the multivariate logistic regression analyses of delayed anastomotic occlusion (table 4), characteristics with P<0.10 in the univariate regression were included in multivariate regression analyses with adjustments for other characteristics. Therefore, only collateral circulation Grade III (OR, 8.192; 95%CI, 2.027-33.105; P=0.003) and preoperative CTP Grade I-II (OR, 4.978; 95%CI, 1.617-15.328; P=0.005) were included in the multivariate logistic regression analyses. Other parameters were not significantly correlated with delayed anastomotic occlusion in univariate regression analysis (P>0.10). In addition, limited to only 19 patients with delayed anastomotic occlusion, up to two variables can be included in the multivariate logistic regression analyses.

According to your suggestion, in order to fully explore the potential independent predictors, we adopt the stepwise forward model and the full factor model to conduct the multivariate logistic regression analysis again, but no other independent predictors are identified.

Change to Text : None.

Comment 3: English should be revised in some sentences: e.g. "Hemorrhagic presentation was less common than intracranial ischemia as the two main manifestations, but it is the leading cause of death in MMD" ; 

Authors’ Response : We thank this reviewer for this suggestion. We have corrected the major grammatical and orthographic typos and errors in the revised manuscript. The manuscript is under language edition as well. Misleading phrasings are revised.

Change to Text : “Hemorrhagic presentation was less common than intracranial ischemia in the manifestation of MMD, but the hematoma is the leading cause of death in MMD”

We correct the sentence in line 44-45.

Comment 4: Lines 256-276: "4.2. Pathogenesis of delayed anastomotic occlusion" Unfortunately, delayed occlusion represents a potential complication in vascular surgery as a whole. This point should be discuss more in the section. Please look at these 3 important references.:     Occurrence and impact of delayed cerebral ischemia after coiling and after clipping in the International Subarachnoid Aneurysm Trial (ISAT). J Neurol. 2012 Apr;259(4):679-83. doi: 10.1007/s00415-011-6243-2.      -      Wrapping of intracranial aneurysms: Single-center series and systematic review of the literature. Br J Neurosurg. 2015;29(6):785-91. doi: 10.3109/02688697.2015.1071320.     -      Delayed sudden hearing recovery after treatment of a large vertebral artery aneurysm causing hearing loss and imbalance: a case report. Br J Neurosurg. 2019 Dec 3:1-5. doi: 10.1080/02688697.2019.1698013. 

Authors’ Response : We thank this reviewer for this suggestion. Indeed, delayed occlusion represents a potential complication in vascular surgery as a whole, vasospasm, space occupying effect, thrombosis, and altered perfusion may be the underlying mechanisms. In this study, the perioperative anastomotic occlusion in hemorrhagic MMD might be caused by thrombogenic event, and the delayed anastomotic occlusion might be caused by the relatively adequate intracranial perfusion compensation.

Change to Text : “Delayed occlusion represents a potential complication in vascular surgery as a whole, vasospasm, space occupying effect, thrombosis, and altered perfusion may be the underlying mechanisms1-3.”

We supplemented this sentence in the line 254-256.

Comment 5: Lines 297-300: "Interestingly, 21.1% of the patients with delayed anastomotic occlusion achieved satisfactory neoangiogenesis...  the brain surface" Can the authors explain this concept better?

Authors’ Response : We thank this reviewer for identifying this issue. Actually, 4 hemispheres (4/19, 21.1%) with delayed anastomotic occlusion were found satisfactory neoangiogenesis in the long-term angiographic follow-up. However, all hemispheres in this study underwent direct revascularization. Then, why did these patients produce satisfactory angiogenesis with anastomotic occlusion? We hypothesized that the underlying mechanism of anastomotic occlusion in these 4 patients may not be caused by relatively adequate intracranial perfusion compensation, but by thrombotic events or compression of the bypass vessels. And the neoangiogenesis was originates from the fascial tissue on the surface of donor vessel close to the brain surface.

Change to Text : Interestingly, 21.1% of the patients with delayed anastomotic occlusion achieved satisfactory neoangiogenesis, the mechanism may be that the donor vascular occlusion is a kind of chronic and progressive occlusion, during which neoangiogenesis was originates from the fascial tissue on the surface of donor vessel close to the brain surface.

We supplemented this sentence in the line 298-299.

References:

  1. Dorhout Mees SM, Kerr RS, Rinkel GJ, Algra A, Molyneux AJ: Occurrence and impact of delayed cerebral ischemia after coiling and after clipping in the International Subarachnoid Aneurysm Trial (ISAT). J Neurol 259:679-683, 2012
  2. Nussbaum ES, Goddard JK, Lowary J, Robinson JM, Hilton C, Nussbaum LA: Delayed sudden hearing recovery after treatment of a large vertebral artery aneurysm causing hearing loss and imbalance: a case report. Br J Neurosurg:1-5, 2019
  3. Perrini P, Montemurro N, Caniglia M, Lazzarotti G, Benedetto N: Wrapping of intracranial aneurysms: Single-center series and systematic review of the literature. Br J Neurosurg 29:785-791, 2015